# Safety and Efficacy of an Innovative Everolimus-Coated Balloon in a Swine Coronary Artery Model

**DOI:** 10.3390/life13102053

**Published:** 2023-10-13

**Authors:** Christos S. Katsouras, Alexandros Tousis, Georgios Vasilagkos, Arsen Semertzioglou, Athanassios Vratimos, Ioanna Samara, Georgia Karanasiou, Vasileios S. Loukas, Grigorios Tsigkas, Dimitrios Fotiadis, Lampros K. Michalis, Periklis Davlouros, Anargyros N. Moulas

**Affiliations:** 12nd Department of Cardiology, University Hospital of Ioannina, University of Ioannina, 45110 Ioannina, Greece; cskats@yahoo.com (C.S.K.); ioan.samara31@gmail.com (I.S.); lamprosmihalis@gmail.com (L.K.M.); 2Department of Cardiology, University Hospital of Patras, 26504 Patras, Greece; alextousis21@gmail.com (A.T.); giorgosvasilagkos@gmail.com (G.V.); gregtsig@upatras.gr (G.T.); pdav@upatras.gr (P.D.); 3Rontis Hellas SA, 41500 Larissa, Greece; arsen.semertzioglou@rontis.com (A.S.); athanassios.vratimos@rontis.com (A.V.); 4Department of Biomedical Research, Institute of Molecular Biology and Biotechnology, Department of Materials Science and Engineering, Unit of Medical Technology and Intelligent Information Systems, University of Ioannina, 45110 Ioannina, Greece; g.karanasiou@gmail.com (G.K.); billloukas@gmail.com (V.S.L.); fotiadis@uoi.gr (D.F.); 5General Department, University of Thessaly, 41500 Larissa, Greece

**Keywords:** drug coated balloon, everolimus, coronary arteries, angioplasty, porcine model

## Abstract

Background: Drug-coated balloons have been used as a non-stenting treatment in coronary and peripheral artery disease. Until recently, only sirolimus- and paclitaxel-coated balloons have been investigated in clinical trials. We evaluated the safety and efficacy of an innovative everolimus-coated balloon (ECB) in a swine coronary artery model. Methods: thirty-two swine coronary arteries were prepared through dilatation with a non-coated angioplasty balloon in a closed-chest model. During a period of 90 days, the following four groups (four animals per group, two coronary arteries per animal) were compared for safety and efficacy: A, Rontis ECB with 2.5 μg/mm^2^ of drug per balloon surface; B, Rontis ECB with 7.5 μg/mm^2^; C, Rontis Europa Ultra bare balloon; and D, Magic Touch, Concept Medical, sirolimus-coated balloon with a drug load of 1.3 μg/mm^2^. Results: Differences in local biological effects (arterial reaction scores) and surface of intimal area (mm^2^) were not statistically significant between the treatment groups. Numerically, group A showed the lowest intimal area and intimal mean thickness, while group B showed the lowest stenosis among all groups. Conclusions: ECB was safe and effective in a porcine coronary artery model. The dose of everolimus may play a role in the biocompatibility of the balloon.

## 1. Introduction

The routine use of drug-eluting stents (DES) for percutaneous treatment of coronary disease has shown that DES are more effective in preventing restenosis than bare-metal stents [1]. However, in-stent-restenosis (ISR) remains a major complication even after new generation DES have been introduced into the market. Recent data has suggested that treatment of ISR may account for 5–10% of all percutaneous coronary procedures performed [2,3]. A third consecutive procedure in recurrent ISR involving a third DES implantation results in a further increase in the ISR rate [4]. An additional drawback of current DES is the need for dual antiplatelet therapy at least four months after angioplasty, whereas a significant proportion of patients must discontinue the administration of the second antiplatelet agent for non-cardiac reasons soon after the procedure.

Drug-coated balloons (DCBs) are endovascular balloons coated with a drug that is delivered to the target endothelial site during inflation and contact with the vascular tissue without leaving a permanent implant behind following the procedure [5]. Commercially available DCBs for coronary interventions use paclitaxel and sirolimus. Studies reported that DCBs are an alternative to DES for ISR [6,7]. Moreover, there is evidence that treatment of de novo coronary lesions with DCBs is associated with a similar risk of restenosis and a lower risk of target lesion thrombosis compared to DES in patients with specific anatomical or clinical characteristics [8]. Furthermore, DCBs offer the option for a shorter duration of double antiplatelet therapy, allowing their use in patients with a high risk for bleeding [8].

Everolimus is a synthetic immunosuppressant derived from a chemical modification of rapamycin that promotes cell cycle arrest in the late G1 phase. Everolimus was originally used in second-generation drug-eluting stents and, due to its proven safety and efficacy, was subsequently used in newer drug-eluting stents and bioresorbable stents [9].

Everolimus has not been tested in DCBs. Based on the fact that everolimus-eluting stents generally perform better than paclitaxel-eluting stents and, in some aspects, better than sirolimus-eluting stents [10,11], we hypothesized that an everolimus-coated balloon could perform better than existing DCBs. The primary scientific questions of this study were whether the developed everolimus balloon is safe and efficient in a swine coronary artery model and what the possible effects of the therapeutic dose of the drug are.

## 2. Materials and Methods

Sixteen female pigs (four animals in each of the four groups, as described below), 5–6 months of age, weighing between 50 and 55 kg, were used. The number of necessary samples was determined with power analysis [12]. The experiments were conducted at a licensed facility of the University Hospital of Patras (University of Patras, Patras, Greece). The facility was equipped with individual cages in climate-controlled rooms. The animals were given a minimum 24-h period for acclimatization after transportation to the facility and experimental procedures. Food was withheld 12 h prior to anesthesia. The test protocols were approved by the Animal Care and Use Board of Patras University Hospital’s ethics committee and by the veterinary board of Western Greece’s Provincial authorities. Additionally, the protocols complied with the medical research ethical principles of the Declaration of Helsinki, the ARRIVE guidelines, and the European Union legislation according to the principle of the 3Rs (Reduction, Refinement, Replacement), ensuring an adequate but low number of animals, improved experimental techniques, and living conditions such that the animals were kept to minimum pain or suffering [13,14].

### 2.1. Anesthesia and Intubation

All animal experiments were performed under general anesthesia (15 mg/kg intramuscular ketamine and 2 mg/kg xylazine) followed by endotracheal intubation (FiO_2_:0.4). Continuous propofol infusion (1–1.4 mg/kg) was administered during anesthesia [15,16]. Pulse oximetry (via the animal’s tail) and heart rate were monitored throughout the operation. Each animal was marked with their assigned number in both ears after anesthesia and prior to the procedure.

The right superficial femoral artery was chosen for arterial access under ultrasound guidance [17]. The Seldinger technique with a 21-gauge needle was used, a 6-Fr-10 cm radial artery sheath (Terumo Medical Corporation^®^, Shibuya, Japan) was inserted, and 100 U/kg heparin was administered intra-arterially. Intra-arterial blood pressure was monitored throughout the procedure. Dual antiplatelet therapy (aspirin 100 mg plus Clopidogrel 75 mg) was administrated to each animal daily for 2 days prior to the procedure and for 10 days afterward, followed by administration of Aspirin 100 mg daily for 80 days.

### 2.2. Coronary Artery Balloon Dilatation

Procedures were performed with a Philips Allura Flat-Panel Angiography Unit (Philips, Amsterdam, The Netherlands). All procedures were carried out by three experienced operators (C.S.K., G.T., and P.D.). Engagement of the left and right coronary arteries was mainly achieved by the Amplatzer AR1^®^ (Cook Medical^®^, Bloomington, IN, USA.) catheter and the Right Coronary Bypass^®^ (RCB) catheter (Cordis Corporation^®^, Hialeah, FL, USA), respectively.

Four groups were compared: group A, Rontis everolimus-coated balloon 2.5 μg/mm^2^ of drug per balloon surface; group B, Rontis everolimus-coated balloon 7.5 μg/mm^2^; group C, Rontis Europa Ultra bare balloon; and group D: sirolimus-coated balloon with a dose of 1.3 μg/mm^2^ (Magic Touch, Concept Medical, Gujarat, India). The test device (everolimus DCB) was a drug-coated angioplasty catheter based on the Europa Ultra coronary balloon catheter delivery platform (Rontis Hellas SA, Larissa, Greece). The balloon’s surface was coated with a proprietary biocompatible excipient system and a therapeutic dose of the drug everolimus (2.5 or 7.5 μg drug per mm^2^ of balloon surface). This formulation forms a homogenous film-like coating around the balloon that helps reduce drug wash-off when advancing the catheter and navigating through the blood vessels and promotes effective drug delivery into the vessel wall when the balloon is deployed at the target lesion site.

Initially, a coronary angiogram and an Optical Coherence Tomography Study (OCT, Dragonfly^TM^ OPTIS^TM^ Imaging Catheter, Abbott, IL, USA) were performed. Afterward, the vessel was prepared through dilatation with a standard, non-coated angioplasty balloon (Europa Ultra, 3.5 mm × 15 mm, Rontis Hellas SA, Larissa, Greece) with a balloon-to-artery diameter ratio of 1.1:1, aiming to cause intimal damage and trigger the onset of vascular repair in order to develop an atherosclerotic lesion (two coronary vessels per animal, one dilatation per vessel). The location of the target sites was specified by fluoroscopy of the radio-opaque markers on the balloon catheters in relation to other landmarks of vascular anatomy, such as the ostium of the treated artery, the left main (or other) bifurcation, diagonal and septal branches, marginal branches, and right artery branches. Usually, the area of interest was located a few millimeters distally or proximally to an arterial bifurcation. Data from the OCT were also used for verification of the lumen’s diameter and as an additional landmark, e.g., the distance from the ostium or other marks. The best angiographic image was selected for use as a visual guide during the second phase of the experiment and was displayed alongside the working monitor.

Then, the angiographic table was locked, and, using the aforementioned angiographic image, the balloon catheters under investigation were deployed. Each pig received two balloon “treatments” in two corresponding arteries (one treatment per artery; either the Left Anterior Descending Artery (LAD), Right Coronary Artery (RCA), and/or Left Circumflex Artery (LCX)). The balloons (3.5 mm × 20 mm in all 4 groups) were inflated at the area of interest. The everolimus DCBs and bare balloons were inflated for 60 s at a pressure of 10 atm, while the sirolimus DCBs were inflated at 8 atm. Inflation pressure was applied in accordance with the specific compliance chart of each product, aiming to overinflate the balloon in order to reach a diameter 0.1–0.2 mm greater than the nominal luminal diameter. OCT was performed after the inflation in order to detect possible dissections or any other problems in the vasculature. Finally, the vascular status was assessed by coronary angiography. A vascular closure device (Angio-Seal 6F, Terumo Medical Corporation^®^, Shibuya, Japan) was used to seal the catheterization site, while an ultrasound verified the absence of local bleeding.

The study duration was 90 days. At the end of the study period, new angiography and OCT studies were performed on the treated vessels, and thereafter, the animals were sacrificed, and specific tissues (the target arterial sites and biopsies from specific internal organs, including the myocardium) were collected for histomorphometry and histopathological evaluation. A macroscopic (visual) investigation of the internal organs also took place, looking for any signs of systemic toxicity possibly associated with DCB deployment and drug/excipient wash-off. Data from OCT were collected for a parallel study aiming to develop an in silico simulation of the process and compare OCT data with data from histomorphometry. For the purposes of this study, data from the OCT were used only for verification of the lumen’s diameter and as an additional landmark of the area of interest.

### 2.3. Histotechnology and Histomorphology by Image Analysis

A scoring system reference was used to assess the arterial wall reaction at the dilatation areas by recording endothelial cell loss, deposits of fibrin attached to the intima, tunica intima proliferation, tunica intima and/or media inflammation, focal or diffuse medial hypertrophy and fibrosis, lamina elastica rupture, smooth muscle proliferation in the intima, proteoglycan/collagen presence and distribution, arterial inflammation, medial smooth muscle cell (SMC) loss, the presence of markers suggesting a host reaction associated with the process (polymorphonuclear cells, lymphocytes, plasma cells, macrophages, giant cells, necrosis, fibrosis, peristrut hemorrhage/fibrin accumulation, neovascularization, fatty infiltrate), elastic lamina (EL) rupture (external EL rupture, internal EL rupture), and medial hypertrophy (focal, diffuse). A scoring system, adapted from ISO 10993-6:2016, was applied. A score from 0 to 4 for each parameter was applied according to the histological findings. Score differences between 0.0 to 2.9 were considered no or minimal host reaction, 3.0 to 8.9 slight host reaction, 9.0 to 15.0 moderate host reaction, and ≥ 15.1 severe host reaction compared to a reference material, as per ISO 10993-6:2016. Limited systemic effects were also evaluated.

Regarding the quantitative parameters analyzed by histopathology, each treatment site was transversally trimmed at five approximately equidistant levels of the artery segment, two End Segments (End 1 and End 2), two Middle-End Segments (Middle-End 1 and Middle-End 2) and one Middle Segment, then dehydrated, embedded in paraffin wax, sectioned at an approximate thickness of 2–4 μm, and stained with Hematoxylin and Eosin (HE) and Elastin Trichrome (ET) (Figure 1).

All arterial segments were image scanned by an Olympus Slideview VS200 slide scanner using a VS-264C camera and 20× objective (Olympus Life Science, Hamburg, Germany). Quantitative evaluation was performed using Olympus imaging and image analysis software cellSens v1.18 (Olympus Life Science, Hamburg, Germany). The following parameters were measured on the two HE-stained end segments (End 1 and End 2) and three HE-stained middle segments (Middle-End 1, Middle, and Middle-End 2) for all arteries: area within external elastic lamina (EEL; μm^2^), area within internal elastic lamina (IEL; μm^2^), lumen (μm^2^), intima (μm^2^, calculation: IEL − Lumen), media (μm^2^, calculation: EEL − IEL), stenosis [%, calculation: 100 − (100 × Lumen/IEL)], and intimal mean thickness (μm) (average value from 10 equidistant thickness measurements) (Figure 2).

Quantitative parameters were overall combined [either End segments (End 1 + End 2), Middle segments (Middle-End 1 + Middle + Middle-End 2), or all segments (End 1 + Middle-End 1 + Middle + Middle-End 2 + End 2)]. For intimal thickness, a mean value was calculated from ten measured values per vessel. These arithmetic mean values were used for further descriptive statistics.

Tissues from the spleen, liver, kidneys, lungs, and myocardium were also dehydrated, paraffin-embedded, sectioned at an approximate thickness of 2–4 μm, and stained with Hematoxylin and Eosin (HE) in order to detect any signs of systemic toxicity occurring due to ECB application. All sections were QC under light microscopy.

### 2.4. Data Comparison and Statistical Analysis

The devices were compared for safety and efficacy. The primary safety endpoint was the absence of major adverse events (death or myocardial infarction) occurring immediately after the intervention and up to three months later. We also recorded any signs of coronary thrombus formation immediately after balloon inflation through angiography. Efficacy endpoints were based on the results of the histology and morphometry. The primary efficacy endpoint referred to the statistically significant comparison of neointimal formation (“intimal area” from the quantitative evaluation of histopathology) and arterial wall reaction score (from histology) between the test groups. Secondary endpoints involved an assessment of the area within the external elastic lamina, the area within the internal elastic lamina, lumen area, intima, media, lumen stenosis, and intimal mean thickness. Regarding systemic effects, findings from the spleen, liver, kidneys, lungs, and myocardium were also recorded. Statistical tests were performed using GraphPad Prism 9 (Graphpad Software, Boston, MA, USA). Descriptive statistics were used for the medial area, intimal area, stenosis, and intimal thickness. The Shapiro-Wilk test for normality was performed. When the data followed a normal distribution, the comparisons were performed with the unpaired Student’s *t*-test. When data did not follow a normal distribution, the Mann-Whitney test was used (in all cases, *p*-values < 0.05 were considered statistically significant).

## 3. Results

The balloons were successfully dilated into the target arteries, and in all cases, no severe complications were detected during the procedure. No sustained or non-sustained ventricular tachycardias were noted, and no severe dissections were seen in the final angiography. No angiographic thrombus was detected. All closure devices were placed uneventfully.

One animal (group C, bare balloons) died; the pig did not wake up after the angioplasty procedure without an obvious cause identified by the veterinarian. All other animals (*N* = 15) survived the scheduled study period, and 30 “treated” coronary arteries were examined.

### 3.1. Systemic Reactions

Thrombosis within the myocardium or systemic organs was not detected. Porcine pleuropneumonia was observed in four animals, a very common condition in young pigs, which usually leads to chronic sequels, such as pleural fibrosis, often with adhesions to the pericardium. Within the kidneys, the presence of minimal focal chronic infarcts in two pigs was noted, one from Group A and one from Group D. There were no significant differences in incidence/severity among the treatments.

The histological evaluation of the LAD, RCA, and LCX myocardial irrigated areas revealed focal perivascular inflammatory infiltrate of small-sized arteries in animal no. P03 (grade 1, group A) and in animal no. P12 (grade 2, group B) of a medium-sized artery with medial hyperplasia. There were no associated changes in the adjacent pericardium, such as necrosis or degeneration. Other findings within LAD, RCA, and LCX myocardial irrigated areas, such as mixed cell inflammatory infiltrate or the presence of myotubes, could be associated with the balloon deployment, but there were no relevant differences in incidence or severity between groups.

### 3.2. Arteries at the Dilatation Site and Host Reaction

The arterial wall reaction at the deployment sites varied in degrees of severity and consisted of one or more of the following findings.

(a)Vascular wall findings: the endothelium lining often showed multifocal loss of endothelial cells. Occasionally, there were multifocal minimal to slight deposits of fibrin attached to the endothelial surface and increased intimal layer thickness due to smooth muscle proliferation and deposition of proteoglycan or collagen production;(b)Arterial inflammation: the tunica intima and/or media from most of the samples presented with minor infiltrate of inflammatory cells;(c)The tunica media appeared thickened by multifocal hypertrophy of the smooth muscle cells;(d)The occasional presence of lamina elastica rupture, more predominantly in the internal lamina;(e)Host reaction associated with the balloon treatment sites consisted of minimal to slight infiltrate in a small number of neutrophils (polymorphonuclear cells), lymphocytes, and/or macrophages.

Fibrosis at the treatment site was minimal in two samples (one from group A and one from group B) and moderate in one sample from group D, where fatty infiltrate was also observed.

Numerically, the least severe host reaction associated with the balloon expansion site was found in groups A and B, followed by groups C and D. However, differences (≤1.4 points) were minor among the treatment groups and per ISO 10993-6:2016 definition (Table 1). Representative pathology images of treated artery segments, one for each treatment group, are shown in Figure 3.

When combining all sections together (End 1, End 2, Midddle-End 1, Middle, and Middle-End 2), group A showed the lowest intimal area and intimal mean thickness, and group B showed the lowest stenosis among all groups. However, Group C showed the lowest medial area. No statistically significant differences were observed between the groups, except for the medial area, for which groups A and B showed a statistically significantly higher medial area than group C (*p* = 0.0054 and *p* = 0.0031, respectively), as seen in Table 2 and Figure 4.

In the Middle sections, group C showed the lowest medial area, intimal area, stenosis, and intimal mean thickness when compared to groups A, B, and D. Groups A and B showed a statistically significantly higher medial area than group C (*p* = 0.0270 and *p* = 0.0160, respectively) (Table 3).

In the End sections, group C showed the highest intimal area, stenosis, and intimal mean thickness when compared to groups A, B, and D. Group A showed the lowest intimal area, stenosis, and intimal mean thickness values among all groups. No statistically significant differences were observed between the groups, except for stenosis, for which group A showed a statistically significantly lower stenosis than group C (*p* = 0.040) (Table 4).

Other findings within the LAD, RCA, and LCX myocardial irrigated areas referring to mixed cell inflammatory infiltrate or the presence of myotubes could be associated with the balloon implantation, but there were no relevant differences in incidence or severity between treatments.

## 4. Discussion

In the current experimental study, we investigated whether or not the use of everolimus DCBs was safe and effective for the treatment of coronary arterial sites where an injury had been caused through the deployment of bare balloons using a swine model. We compared Rontis’ everolimus DCBs (2 doses) with a non-coated balloon and a commercially available DCB. The results suggested that: (1) the devices were safe; (2) the test balloons created the least severe host reaction, although differences among all groups were not statistically significant; (3) their use was associated with minimal neointimal formation, especially at the ends of “injury” areas produced by plain balloons (less reaction with 2.5 μg/mm^2^ balloon), although the differences were not statistically significant; and (4) the dose of everolimus may play a role in the biocompatibility of the device.

Everolimus, a semi-synthetic derivative of naturally occurring rapamycin with potent immunosuppressive and anti-proliferative effects, is one of the four mammalian targets of rapamycin (mTOR) inhibitors: sirolimus, everolimus, temsirolimus, and ridaforolimus [18,19]. These large molecules (molecular weight ~1000 kDa) inhibit the action of the mTOR protein kinase complex through the binding of the FK506 binding protein-12 (FKB12), which forms a ternary complex with mTOR [20].

Everolimus reduced neointimal proliferation in cultured human saphenous vein grafts [21]. Moreover, stent-mediated delivery of everolimus inhibited the formation of neointimal hyperplasia and neoatherosclerosis in porcine iliac arteries [22]. Εverolimus eluting stents (EES) have been widely utilized in clinical practice in patients undergoing percutaneous coronary interventions. In a large randomized trial, the incidence of target lesion failure and stent thrombosis at 12 months post-intervention were similar in patients treated with either EES or sirolimus-eluting stents (both stents with durable polymers) [23]. However, at five years, the rates of target lesion revascularization, target vessel revascularization, recurrent myocardial infarction, and stent thrombosis were significantly lower in the EES group compared to the SES group [24]. Moreover, only the use of SES with a biodegradable polymer and/or ultrathin struts resulted in similar results to the use of EES with a durable polymer and thicker struts [25,26,27]. The causes for this “advantage” of everolimus are not clear. Minor differences in structure, systemic clearance, vessel wall levels of the drug, and the relative hydrophobicity of sirolimus may play a role [18,19].

DCBs have been used to treat ISR and de novo coronary disease with promising results. Different DCBs may result in differences in efficacy (late lumen loss) [28]. No everolimus DCB has been tested in experimental models (or in clinical practice). We investigated the results of a novel everolimus DCB in an experimental model of injury created by plain balloons in non-atherosclerotic swine coronary arteries. We tested two different doses of everolimus (2.5 or 7.5 μg drug per mm^2^ of balloon surface). In the comparative evaluation (with a non-coated balloon and sirolimus DCB), differences were limited among the groups in total arterial reaction average score. Numerically, a lower score was found in groups with everolimus DCBs (both doses). There was no fibrosis at the treatment site in most cases (fibrosis was minimal in two samples and moderate in one). To the best of our knowledge, there are no similar studies for comparison. One study examined an everolimus-eluting bioresorbable vascular scaffold (Absorb) and the second-generation everolimus-eluting cobalt chromium XIENCE V stent in a porcine coronary artery model and found that inflammation was mild at six months in both groups. The inflammation score was greater at 12, 18, 30, and 36 months post-intervention in the Absorb and XIENCE groups. Both devices exhibited absent or minimal inflammation at 42 months [29]. In our DCB study, we provided data at only 90 days post-intervention. It is well known that increased inflammation has been correlated with a greater neointimal thickness and ISR in bare metal stents and DESs [30,31]. However, “continuous inflammation” is an important factor in the ISR, while the role of inflammation in restenosis after balloon angioplasty does not have the same significance as in ISR [32].

The intimal area was lower in everolimus-DCB groups, especially at the ends of “areas of interest”. It is interesting that remodeling and neointima formation at the proximal reference segments significantly affect the restenotic process after successful plain balloon coronary angioplasty in humans [33]. The small differences among the groups may be due to the limited number of treated arteries and the observed large standard deviations for all groups and all parameters (data in Supplement). However, based on the results of both the intimal area and inflammatory response, we can assume the everolimus DCB is at least equivalent in safety and efficacy to the commercially available sirolimus-DCB. It is also interesting that we recorded small differences in the two groups of everolimus DCBs (2.5 or 7.5 μg drug per mm^2^ of balloon surface). Histomorphometrically, the media area was greater in groups A and B compared to group C in the Mid-Segments (not in the End-Segments). However, groups A and B had a numerically larger EEL area, IEL area, and lumen area than group C in the same Mid-Areas. These results may be the consequences of positive remodeling after everolimus DCBs, as in an everolimus-eluting bioresorbable vascular scaffold [29], and, in combination with a lower intima area, indicate that the media area differences were not deemed to induce more stenosis in groups A and B.

### Limitations

Our study has some limitations. First, vascular responses to the balloons in healthy (without atherosclerotic lesions) swine coronary arteries are likely different from those in diseased human arteries. Second, the observation period was three months. We do not know if we would have obtained the same results at different/additional time points after the procedure. Third, we performed the experiments on pigs weighing 50–55 kg, whereas the same animals, three months later, weighed more than 80 kg. Finally, despite all efforts to mark the “areas of interest” accurately, small divergences in balloon positioning between the first and second dilatation could not be excluded.

More than 25 years have passed since researchers started to examine local drug delivery for arteriosclerotic lesions [34]. During the last decade, the proportion of patients (and lesions) who underwent DCB coronary (or peripheral) angioplasty in catheterization laboratories has increased significantly [35]. Different drugs in DCB devices have not been tested in large, randomized, clinical head-to-head comparison studies despite the fact that experimental data suggest the absence of a class effect even within DCBs with the same drug but differences in excipients and catheter properties [36,37]. There is only clinical data from a study that evaluated two different paclitaxel DCBs for in-stent restenosis, a small study comparing paclitaxel- versus sirolimus-DCBs in de novo coronary lesions, and from indirect comparison between paclitaxel- and sirolimus-coated balloons exist [28,38,39]. Our experimental data showed that an everolimus-coated balloon device is feasible and at least equivalent in safety and efficacy to a commercially available sirolimus-coated balloon. We think that accumulated experimental and clinical data are necessary not only to promote the conduction of randomized clinical trials testing different drugs between different DCBs but also to better understand the “response to injury” cataract in atherosclerotic lesions, de novo, or within a previously placed stent. Maybe, in the end, every DCB device will find a different place on the self of the catheterization laboratory. We think that we are in the middle of the road. Obviously, further research is needed.

## 5. Conclusions

The present research is the first to investigate the safety and efficacy of an innovative everolimus DCB in an experimental model of swine coronary arteries. The use of an everolimus DCB is safe and effective with an acceptable degree of neointimal thickness 90 days post-intervention. The dose of everolimus may play a role in the biocompatibility of the balloon.

## Figures and Tables

**Figure 1 life-13-02053-f001:**
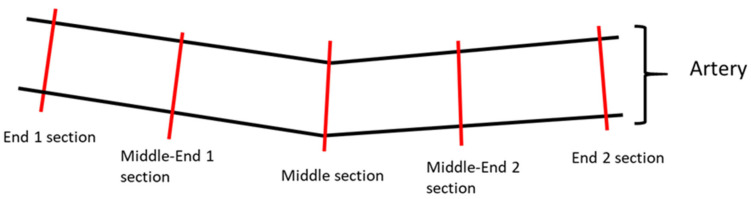
Illustrative image of the sectioning of the artery samples. Each treatment site was transversally trimmed at five approximately equidistant levels of the artery segment: two end sections, two middle-end sections, and one middle section.

**Figure 2 life-13-02053-f002:**
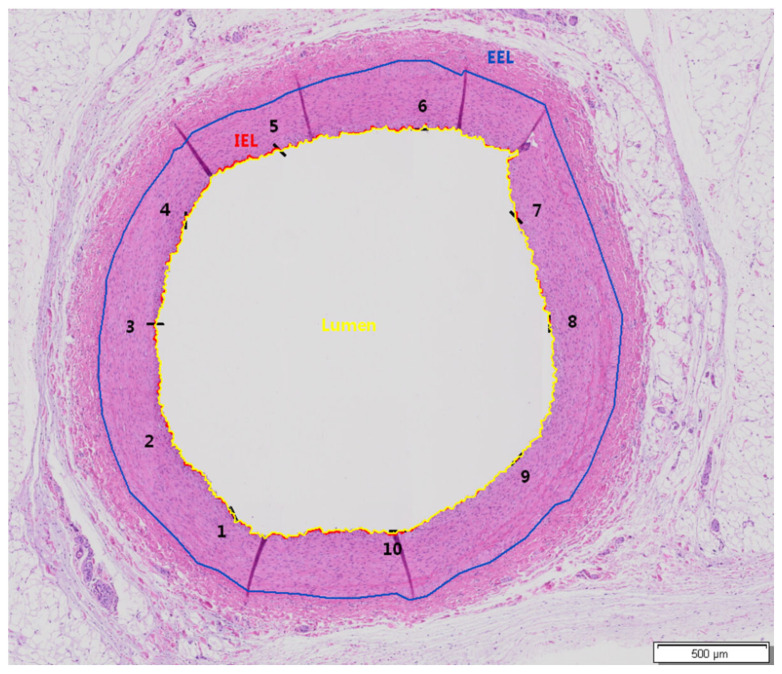
Exemplary image indicating the measured parameters in the examined artery segments. Animal 16, Middle-End 1 Segment, Artery RCA, group A, HE, and objective ×20. External elastic lamina (EEL) (blue color), internal elastic lamina (IEL) (red color), and lumen (in yellow). Intimal thickness measurements were performed at ten equidistant points (black color), and the average was calculated. The numbers (1–10) indicate the positions where thickness measurements were conducted.

**Figure 3 life-13-02053-f003:**
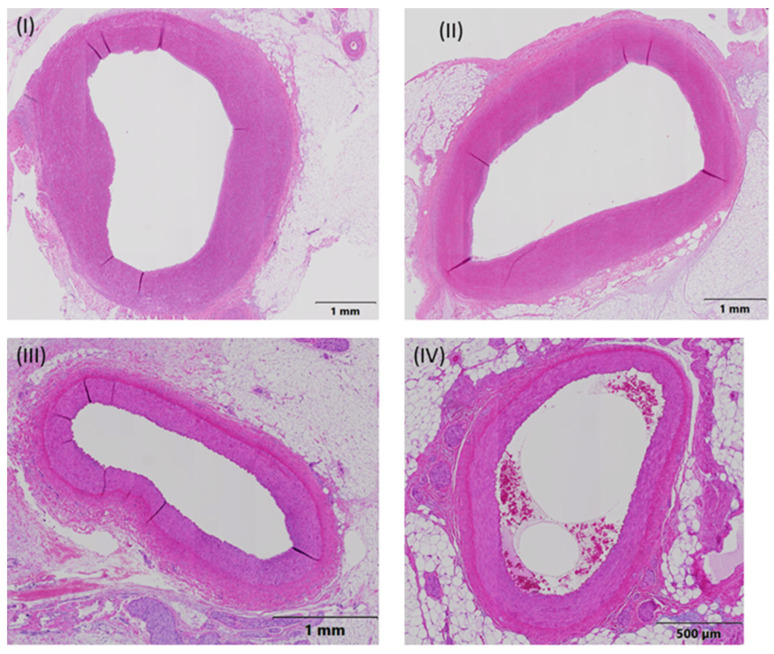
Representative pathology images of treated artery segments, one for each treatment group. The measurement parameters were the external elastic lamina (EEL), internal elastic lamina (IEL), lumen, intima, and media. Ten approximately equidistant measurements were used to measure the intimal thickness. (**I**) Animal 03, RCA, Middle-End 2, group A (everolimus-coated balloon 2.5 μg/mm^2^), (**II**) Animal 04, LAD, Middle-End 1, group B (everolimus-coated balloon 7.5 μg/mm^2^), (**III**) Animal 05, LCX, Middle-End 1, group C (uncoated (bare) balloon), and (**IV**) Animal 06, LAD-End 1, group D (sirolimus-coated balloon 1.3 μg/mm^2^).

**Figure 4 life-13-02053-f004:**
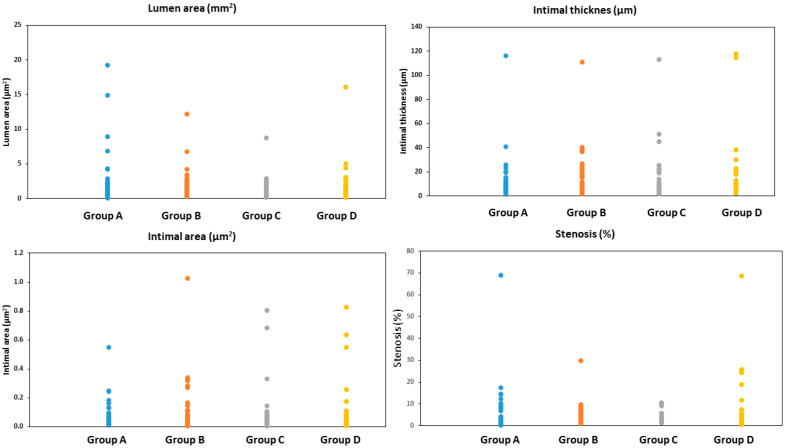
The lumen area, intimal mean thickness, intimal area, and % stenosis in all treated arterial sections combined (End 1, End 2, Midddle-End 1, Middle, and Middle-End 2). Group A = everolimus-coated balloon 2.5 μg/mm^2^, group B = everolimus-coated balloon 7.5 μg/mm^2^, group C = uncoated (bare) balloon, and group D = sirolimus-coated balloon 1.3 μg/mm^2^.

**Table 1 life-13-02053-t001:** Histology results at the balloon deployment sites. Average scores at the arterial balloon deployment sites.

	Group A	Group B	Group C	Group D
Number of animals	4	3	4	4
Total Sample number	26	30	29	34
Sum Host Reaction Score Total	51	53	74	76
Host Reaction Average Score associated with the balloon deployment site	2.0	1.8	2.6	2.2
Vascular wall findings	5.2	5.9	5.7	5.4
Artery Inflammation	1.4	1.3	1.4	1.2
Medial smooth muscle cell (SMC) loss	0.0	0.0	0.0	0.0
Medial smooth muscle cell replacement tissue	0.0	0.0	0.0	0.0
Medial hypertrophy	1.8	2.1	1.6	2.8
Lamina elastic rupture	0.6	0.9	1.0	0.7
TOTAL Arterial Reaction Average Score	11.0	11.9	12.2	12.4

**Table 2 life-13-02053-t002:** The medial area, lumen area, intimal area, intimal mean thickness, and % stenosis (average (SD) with 95% confidence interval (CI)) in all treated arterial sections combined (End 1, End 2, Midddle-End 1, Middle, and Middle-End 2). * *p* = 0.0054, ^#^
*p* = 0.0031 comparison of groups with the respective symbol with group C.

	Group A	Group B	Group C	Group D
Lumen Area (mm^2^)	3.23 (4.58) (1.38–5.08)	2.21 (2.38) (1.29–3.14)	1.64 (1.58) (1.03–2.25)	2.06 (2.82) (1.02–3.09)
Medial Area (mm^2^)	4.39 (6.01) * (1.97–6.82)	3.25 (4.21) ^#^ (1.62–4.88)	1.68 (2.07) (0.87–2.48)	2.49 (3.71) (1.13–3.85)
Intimal Area (mm^2^)	0.098 (0.113) (0.052–0.14)	0.138 (0.202) (0.06–2.16)	0.114 (0.189) (0.041–0.187)	0.114 (0.195) (0.042–0.185)
Stenosis (%)	7.97 (13.4) (2.58–13.37)	5.63 (5.29) (3.58–7.68)	6.41 (8.36) (3.17–9.66)	7.11 (13.12) (2.30–11.92)
Intimal Mean Thickness (µm)	15.06 (22.4)	17.72 (21.2)	17.1 (22.42)	17.36 (27.65)

**Table 3 life-13-02053-t003:** The medial area, lumen area, intimal area, intimal mean thickness, and % stenosis (average (SD, 95% confidence interval)) in the middle sections combined (Midddle-End 1, Middle, and Middle-End 2).

	Group A	Group B	Group C	Group D
Lumen Area (mm^2^)	1.71 (1.76) (0.73–2.68)	1.76 (1.48) (1.00–2.52)	1.47 (0.77) (1.07–1.86)	1.71 (1.30) (1.08–2.33)
Medial Area (mm^2^)	2.61 (2.69) (1.13–4.10)	2.42 (1.91) (1.44–3.41)	1.31 (0.51) (1.05–1.57)	1.94 (2.09) (0.93–2.94)
Intimal Area (mm^2^)	0.11 (0.14) (0.03–0.18)	0.14 (0.25) (0.06–0.19)	0.05 (0.03) (0.04–0.07)	0.13 (0.22) (0.02–0.23)
Stenosis (%)	11.01 (16.88) (1.67–20.35)	6.11 (6.58) (2.73–9.50)	3.97 (3.05) (2.41–5.54)	8.09 (15.67) (0.54–15.64)
Intimal Mean Thickness (µm)	19.2 (28.7) (1.7–20.4)	18.5 (25.7) (2.7–9.5)	10.0 (7.7) (2.4–5.5)	20.7 (34.1) (0.5–15.6)

**Table 4 life-13-02053-t004:** The medial area, lumen area, intimal area, intimal mean thickness, and % stenosis (average (SD) (95% confidence interval) in the end sections combined (End 1, End 2).

	Group A	Group B	Group C	Group D
Lumen Area (mm^2^)	5.30 (6.31) (1.06–9.54)	2.91 (3.30) (0.69–5.12)	1.91 (2.39) (0.30–3.51)	2.60 (4.29) (0.00–5.32)
Medial Area (mm^2^)	6.82 (8.30) (1.24–12.4)	4.53 (6.24) (0.33–8.72)	2.25 (3.26) (0.58–4.43)	3.37 (5.38) (0.00–6.78)
Intimal Area (mm^2^)	0.09 (0.08) (0.04–0.13)	0.13 (0.12) (0.05–0.21)	0.21 (0.28) (0.03–0.4)	0.09 (0.15) (0.00–0.04)
Stenosis (%)	3.80 (3.83) (1.26–6.41)	4.87 (2.27) (3.34–6.39)	10.19 (12.17) (2.01–18.36)	5.57 (7.98) (0.49–10.64)
Intimal Mean Thickness (µm)	9.5 (6.3) (5.3–13.8)	16.5 (12.3) (8.2–24.8)	28.1 (32.3) (6.4–49.8)	12.1 (11.3) (4.9–19.3)

## Data Availability

The data presented in this study are available on request from the corresponding author.

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
