# Peer review of "Safety and Efficacy of an Innovative Everolimus-Coated Balloon in a Swine Coronary Artery Model"

_life, 2023, doi:10.3390/life13102053_

Round 1
Reviewer 1 Report
The present manuscript compares the safety and efficacy of a new everolimus-coated balloon in a closed-chest pig model. The experiments are well conducted and documented, the manuscript is well written and easy to follow. In a formal sense, I don't have any criticisms, except that I miss a meaningful scientific question and I see this article more as a test report for a new device. Additionally, the authors should consider to report single data points rather than bars with standard deviation, see Figure 4.
Author Response
Reviewer 1:
The present manuscript compares the safety and efficacy of a new everolimus-coated balloon in a closed-chest pig model. The experiments are well conducted and documented, the manuscript is well written and easy to follow. In a formal sense, I don't have any criticisms, except that I miss a meaningful scientific question and I see this article more as a test report for a new device.
We thank the reviewer for the meaningful comments. We changed the last paragraph of the introduction and added two more references, as follows:
“Everolimus has not been tested in DCBs. Based on the fact that everolimus eluting stents perform generally better than paclitaxel eluting stents and in some aspects even than sirolimus eluting stents (References: https://pubmed.ncbi.nlm.nih.gov/26860585/, https://pubmed.ncbi.nlm.nih.gov/23351828/) we hypothesized that an everolimus coated balloon could possibly perform better than existing DCBs. The primary scientific questions of this study were whether the developed everolimus balloon is safe and efficient in a swine coronary artery model and what are the possible effects of the therapeutic dose of the drug”. The aim of this study was to evaluate the safety and efficacy of an innovative everolimus-coated balloon in a swine coronary artery model.
Additionally, the authors should consider to report single data points rather than bars with standard deviation, see Figure 4.
We thank the reviewer for the meaningful comments.
In response, we have changed Figure 4 to include single data points.

Reviewer 2 Report
This study wants to show the safety and efficacy of an innovative everolimus-coated balloon (ECB) in a swine coronary artery model. The results show that differences in local biological effects (arterial reaction scores) and surface of intimal area (mm2) were not statistically significant between the treatment groups. Numerically, Group A showed the lowest intimal area and intimal mean thickness, while Group B showed the lowest stenosis among all groups.
In the current experimental study, the author investigated whether or not the use of everolimus DCBs is safe and effective for the treatment of coronary arterial sites where the injury had been caused through the deployment of bare balloons, using a swine model.
The present research is the first to investigate the safety and efficacy of an innovative everolimus DCB in an experimental model of swine coronary arteries. The use of Everolimus DCB is safe and effective with an acceptable degree of neointimal thickness 90 days post-intervention. The dose of everolimus may play a role in the biocompatibility of the balloon. This study is critical because it evaluated the possibility of using sirolimus-medicated stents in an animal model and assessed the risks and benefits.
It is well elaborated although I would recommend a complete English review. But the data are very meager and not statistically significant. I therefore do not consider it suitable for this journal.
This study wants to show the safety and efficacy of an innovative everolimus-coated balloon (ECB) in a swine coronary artery model. The results show that differences in local biological effects (arterial reaction scores) and surface of intimal area (mm2) were not statistically significant between the treatment groups. Numerically, Group A showed the lowest intimal area and intimal mean thickness, while Group B showed the lowest stenosis among all groups.
In the current experimental study, the author investigated whether or not the use of everolimus DCBs is safe and effective for the treatment of coronary arterial sites where the injury had been caused through the deployment of bare balloons, using a swine model.
The present research is the first to investigate the safety and efficacy of an innovative everolimus DCB in an experimental model of swine coronary arteries. The use of Everolimus DCB is safe and effective with an acceptable degree of neointimal thickness 90 days post-intervention. The dose of everolimus may play a role in the biocompatibility of the balloon. This study is critical because it evaluated the possibility of using sirolimus-medicated stents in an animal model and assessed the risks and benefits.
It is well elaborated although I would recommend a complete English review. But the data are very meager and not statistically significant. I therefore do not consider it suitable for this journal.
Author Response
Reviewer 2:
This study wants to show the safety and efficacy of an innovative everolimus-coated balloon (ECB) in a swine coronary artery model. The results show that differences in local biological effects (arterial reaction scores) and surface of intimal area (mm2) were not statistically significant between the treatment groups. Numerically, Group A showed the lowest intimal area and intimal mean thickness, while Group B showed the lowest stenosis among all groups.
In the current experimental study, the author investigated whether or not the use of everolimus DCBs is safe and effective for the treatment of coronary arterial sites where the injury had been caused through the deployment of bare balloons, using a swine model.
The present research is the first to investigate the safety and efficacy of an innovative everolimus DCB in an experimental model of swine coronary arteries. The use of Everolimus DCB is safe and effective with an acceptable degree of neointimal thickness 90 days post-intervention. The dose of everolimus may play a role in the biocompatibility of the balloon. This study is critical because it evaluated the possibility of using sirolimus-medicated stents in an animal model and assessed the risks and benefits.
It is well elaborated although I would recommend a complete English review. But the data are very meager and not statistically significant. I therefore do not consider it suitable for this journal.
We thank the reviewer for their comments. A complete English review has been conducted.
Reviewer 3 Report
- The paper provides a clear background on the use of drug-coated balloons (DCBs) in coronary and peripheral artery disease.
-
It's important to address the sample size in the study. While authors mention 32 swine coronary arteries, it might be helpful to justify why this number was chosen or whether any power analysis was performed (if necessary).
-
The paper mentions that differences in local biological effects and intimal area were not statistically significant. It would be beneficial to include p-values or confidence intervals to support this claim and provide a better understanding of the data.
- The paper mentions that previous clinical trials primarily investigated sirolimus- and paclitaxel-coated balloons. It might be useful to briefly discuss how the results of this study compare or contrast with those trials, as this would provide context for the significance of the findings.
- It might be useful to include the 95% CI in table 2 to table 4.
- Overall, the studies are good, the dose of everolimus may play a role in the biocompatibility of the balloon.
Author Response
Reviewer 3:
The paper provides a clear background on the use of drug-coated balloons (DCBs) in coronary and peripheral artery disease.
We thank the reviewer for this comment.
It's important to address the sample size in the study. While authors mention 32 swine coronary arteries, it might be helpful to justify why this number was chosen or whether any power analysis was performed (if necessary).
We based our sample size calculation on the expected % stenosis of the arteries. Based on literature, we anticipated a stenosis in the range within 6-9%. We hypothesized a difference of 50% stenosis between the highest and lowest group, with the highest 9% and lowest 6 with an SD=2 (33% of the lowest value i.e. 6% stenosis). Power calculation with an alpha 0.05 and power 80% gives a number of 7 samples per group. Since we use two arteries per animal this gives a number of 7/2=3.5 animals. So we used 4 animals.
We added the following phrase in the second line of “Materials and methods”: “The number of necessary samples was determined with power analysis.” and added the following reference: Charan, J., & Kantharia, N. D. (2013). How to calculate sample size in animal studies?. Journal of pharmacology & pharmacotherapeutics, 4(4), 303–306. https://doi.org/10.4103/0976-500X.119726
The paper mentions that differences in local biological effects and intimal area were not statistically significant. It would be beneficial to include p-values or confidence intervals to support this claim and provide a better understanding of the data.
We thank the reviewer for this comment. We added the confidence intervals in tables 2-4.
The paper mentions that previous clinical trials primarily investigated sirolimus- and paclitaxel-coated balloons. It might be useful to briefly discuss how the results of this study compare or contrast with those trials, as this would provide context for the significance of the findings.
We thank the reviewer for this insightful comment. In response, we added the following paragraph at the end of the discussion:
More than 25 years have passed since researchers started to examine the local drug delivery for arteriosclerotic lesions[34]. During the last decade, the proportion of patients (and lesions) who underwent DCB coronary (or peripheral) angioplasty in the catheterization laboratories has been increased significantly[35]. Different drugs in DCB devices have not been tested in large randomized clinical head-to-head comparison studies despite the fact that experimental data suggest the absence of a class effect even within DCB with the same drug but differences in excipients and catheter properties[36, 37]. Only little clinical data from a study evaluated two different paclitaxel DCBs for in-stent restenosis, a small study comparing paclitaxel- versus sirolimus-DCB in de novo coronary lesions and clinical data from indirect comparison between paclitaxel- and sirolimus-coated balloons exist[28, 38, 39]. Our experimental data showed that everolimus-coated balloon device is feasible and at least equivalent in safety and efficacy with a commercially available sirolimus-coated balloon. We think that accumulated experimental and clinical data are necessary not only to promote the conduction of randomized clinical trials testing different drugs between different DCBs, but also to better understand the “response to injury” cataract in atherosclerotic lesions, de novo or within a previously placed stent. Maybe, in the end, every DCB device will find a different place on the self of the catheterization laboratory. We think that we are in the middle of the road. Obviously, further research is needed.
It might be useful to include the 95% CI in table 2 to table 4.
We thank the reviewer for this comment. We added 95% CI in tables 2-4.
Overall, the studies are good, the dose of everolimus may play a role in the biocompatibility of the balloon.
We thank the reviewer for this comment.

Round 2
Reviewer 2 Report
Good Job, this article is suitable for publication in this journal